# Norway spruce shows stronger growth sensitivity and weaker intrinsic water-use efficiency response than Scots pine under increasing water limitation in southern Finland

5 Paul Szejner<sup>1,\*</sup>, Eduardo Martínez-García<sup>1</sup>, Helena Haakana<sup>1</sup>, Juha Heikkinen<sup>1</sup>, Katja T. Rinne-Garmston<sup>1</sup>, Petri Kilpeläinen<sup>1</sup>, Giles Young<sup>1</sup>, Elina Sahlstedt<sup>1</sup>, Raisa Mäkipää<sup>1</sup>, Aleksi Lehtonen<sup>1</sup>

<sup>1</sup> Natural Resources Institute Finland (Luke), Latokartanonkaari 9, FI-00790, Helsinki, Finland

Correspondence to: Paul Szejner, e-mail: paul.szejner@luke.fi

Abstract. Boreal forests, essential for carbon sequestration and multiple ecosystem services, face increasing pressure from climate-induced water stress. This study investigates how increasing water limitation affects growth and intrinsic water-use efficiency (iWUE) in Scots pine (Pinus sylvestris L) and Norway spruce (Picea abies (L.) Karst) in southern Finland. We combined site-level tree-ring data on basal area increment (BAI) and carbon isotope discrimination ( $\Delta^{13}$ C) from three sites per species, representing contrasting soil moisture conditions (dry versus wet), with regional growth indices from the Finnish National Forest Inventory (NFI) spanning 1990-2022. Our results show that forests in southern Finland have become increasingly water-limited over the past decade. Site-level and NFI based growth decline post-2015 is pronounced in Norway spruce, indicating strong sensitivity to water limitation, while Scots pine exhibits only marginal reductions beginning around 2010. Δ<sup>13</sup>C analyses indicate increased stomatal regulation in Scots pine and, to a lesser extent, in Norway spruce after 2015, consistent with intensifying water limitations. iWUE derived from tree ring  $\Delta^{13}$ C increased more steeply Scots pine than in Norway spruce, suggesting weaker physiological adjustment in spruce to rising atmospheric moisture demand. Interannual variability in both growth and iWUE for both species was strongly correlated with the standardized precipitation-evapotranspiration index (SPEI) and vapor-pressure-deficit (VPD). Linear mixed-effects models confirm that Norway spruce growth sensitivity to VPD and SPEI intensified after 2015, whereas Scots pine showed consistent  $\Delta^{13}$ C responses and relatively buffered growth. These findings highlight the growing vulnerability of boreal conifers, particularly Norway spruce, to intensifying water stress. Sensitivity varied by soil type: Scots pine was more responsive on organic soils, while Norway spruce was more vulnerable on mineral soils. Species- and site-specific differences in water-use strategies underscore the importance of adaptive forest management, including species choice, site matching, and silvicultural planning, to support forest resilience and productivity under warmer, drier climate.

#### 1. Introduction

Boreal forests cover approximately one-third of the global forest area and play a crucial role in maintaining ecosystem services and acting as significant carbon sinks (Bonan, 2008; Bradshaw et al., 2009; Gauthier et al., 2015; Pan et al., 2011; Tagesson et al., 2020). However, climate change has intensified climate-induced stressors by increasing temperatures and atmospheric water demand, threatening forest health by reducing tree growth and carbon sink, increasing susceptibility to pests and diseases, and elevating mortality rates (Allen et al., 2010; Girardin et al., 2016; 2023; Junttila et al., 2024; Linnakoski et al., 2017b, a; Martínez-García et al., 2024; Seidl et al., 2017; Zhang et al., 2024). Among these stressors, increasing water limitation has emerged as a fundamental driver of forest growth decline in the boreal region over recent

decades (Gao et al., 2016; Gutierrez Lopez et al., 2021; Treydte et al., 2023). However, species-specific climate effects on growth and water-use regulation remain unclear.

Current research highlights a growing trend in the frequency, duration, and intensity of drought events across Europe (Gutierrez Lopez et al., 2021; Knutzen et al., 2025; Lehtonen and Pirinen, 2019), with pronounced impacts emerging across the boreal region (Gao et al., 2016; Schär et al., 2004; Schuldt et al., 2020). These droughts, often coinciding with elevated air temperatures (Ta), increase vapor pressure deficit (VPD) and reduce soil moisture, are likely to become more frequent under continued warming. (Gao et al., 2016; Swain et al., 2025). In southern Finland, characterized by cool and moist growing conditions, the recent rise in summer Ta has led to elevated atmospheric water demand, which amplify vegetation water stress (Grossiord et al., 2020; Novick et al., 2024; Treydte et al., 2023). Although summer precipitation has shown a modest upward trend during summer in parts of this boreal zone, these gains on precipitation are minor relative to the rising evaporative demand (Grossiord et al., 2020; Irannezhad et al., 2014). As a result, the region continues to experience a sequence of unusual warm and dry years (Knutzen et al., 2025).

In southern Finland, records from the 13<sup>th</sup> Finnish National Forest Inventory (NFI-13) have documented declines in tree growth, increased canopy dieback, and elevated mortality over the past decade, consistent with increasing water limitation. The two dominant tree species, Scots pine (*Pinus sylvestris L.*) and Norway spruce (*Picea abies* (L.) Karst.), which account for approximately 40% and 38% of the total stem volume, respectively, have both grown below expected trajectories, with Norway spruce exhibiting larger negative deviations (H. M. Henttonen et al., 2024; Junttila et al., 2024; K. T. Korhonen et al., 2024; Mäkinen et al., 2022). (see Figure S1 for details on stock across the 11<sup>th</sup> to 13<sup>th</sup> NFIs in southern Finland). However, these large-scale assessments have not directly attributed the growth declines to specific climatic stressors or examined concurrent changes in species-specific physiological responses. This lack of attribution highlights the need for regionally focused, integrated physiological analyses to better understand water limitation impacts and to support adaptive forest management under changing climate conditions.

Tree species-specific traits shape forest responses to water limitation and warming, influencing composition and resilience (Anderegg et al., 2018). In boreal forests, Scots pine and Norway spruce exhibit distinct physiological and water-use strategies under varying water limitations, reflecting their respective strategies for coping with drought (Levesque et al., 2013; Oren et al., 1999). Relative to pine, spruce usually has a shallower and less plastic root system, which increases its sensitivity to short-term fluctuations in water supply (Kalliokoski et al., 2008; Schmid and Kazda, 2002). These species also differ in photosynthetic capacity and stomatal control. Across comparable conditions, pine generally shows higher photosynthetic capacity than spruce (Thum et al., 2008). At the canopy level, Scots pine exhibits a steeper stomatal conductance response with increasing VPD, whereas spruce is less sensitive (Lagergren and Lindroth, 2002). Under frequent high-VPD events, relatively weaker stomatal down-regulation in spruce can lead to increased water loss, lower leaf water potentials, slower cell expansion, and heightened hydraulic risk when water supply is limited (Arend et al., 2021). These differences align with observed patterns of lower drought tolerance in spruce, often seen as more substantial growth reductions under water limitation (Motte et al., 2023; Ulrich and Grossiord, 2023). As water limitation intensifies in southern

100

Finland, resolving such sensitivities among species is critical for anticipating shifts in forest composition and for guiding adaptive management.

Intrinsic water-use efficiency (iWUE), defined as the ratio of net photosynthetic rate (A) to stomatal conductance (gs), reflects a species ability to balance carbon gain and water loss (Farquhar et al., 1989a; Seibt et al., 2008). Differences in iWUE among species can signal variations in stomatal control, photosynthetic capacity, and water-use strategies affecting competitive interactions, and thereby influencing forest structure and dynamics (Ainsworth and Rogers, 2007; Anderegg et al., 2018). For instance, increases in iWUE above the expected increase due to increasing atmospheric CO<sub>2</sub>, generally indicate stronger stomatal regulation in response to water stress (Farquhar et al., 1982, 1989b; Saurer et al., 2004; Szejner et al., 2018; Voelker et al., 2016). iWUE is closely related to  $\delta^{13}$ C because photosynthetic carbon isotope discrimination ( $\Delta^{13}$ C) primarily reflects the ratio of internal-to-ambient CO<sub>2</sub> concentrations ( $c_i/c_a$ ). Since A can be expressed as the product of gs and the CO<sub>2</sub> gradient ( $c_a-c_i$ ), iWUE increases as  $c_i/c_a$  decreases, leading to reduced  $\Delta^{13}$ C values. In some cases, tree species may exhibit long-term stable iWUE, reflected by a consistent increase in  $\Delta^{13}$ C associated with a proportional increase in ( $c_i/c_a$ ) under increasing  $c_a$  (Belmecheri et al., 2021; Saurer et al., 2004; Voelker et al., 2016). Forest management such as thinning and fertilization can influence iWUE by enhancing photosynthetic capacity and resource availability, often resulting in increased A and higher iWUE (Brooks and Mitchell, 2011; Lehtonen et al., 2023a; Palvi et al., 2025). However, increasing iWUE does not always correspond to increased tree growth or productivity, particularly in unmanaged forests where other limiting factors may prevail (Andreu-Hayles et al., 2011; Battipaglia et al., 2013; Walker et al., 2020).

Tree responses to water limitation can show differences between organic and mineral soils because of contrasts in water holding, aeration, and nutrient supply. Specifically, in organic peat soils, high water retention but low oxygen in the rooting zone constrains root function. Drainage-induced lowering the water table (WT) can improve aeration and promote growth but may increase drought exposure if the WT declines too deeply (Hökkä et al., 2025; Sikström and Hökkä, 2016). In peatlands, fine-root production is concentrated in the upper 20 cm, making trees particularly sensitive to WT declines during dry periods (He et al., 2023). This effect may be more significant in well-drained peatlands, where lower WT levels can more readily induce drought stress. In contrast, mineral soils typically drain faster and retain less water than organic soils, causing surface layers to dry rapidly during warm summers, though deeper rooting is possible where soil structure and oxygen availability permit (Heiskanen and Mäkitalo, 2002; Schenk and Jackson, 2002). Topography further modulates these patterns observed in mineral soils. In particular, lowlands tend to retain moisture and risk waterlogging, whereas uplands dry more quickly and face higher evaporative demand, creating microclimatic conditions that influence tree species performance (Kemppinen et al., 2023; McNichol et al., 2024). Understanding these tree-soil interactions is essential for assessing drought sensitivity and developing targeted forest management strategies.

This study focuses on the southernmost region of Finland, which is the most vulnerable to water limitation in the country (Lehtonen and Pirinen, 2019) (Figure 1). Here, we investigate how increasing water limitations influence growth and iWUE of Scots pine and Norway Spruce across mineral and organic soils under contrasting water-availability conditions in this region. To address this, we integrated site-level tree-ring analyses of basal area increment (BAI),  $\Delta^{13}$ C, and iWUE with

regional growth indices (GIs) derived from the Finnish NFI data spanning 1990–2022. We related these growth and physiological responses to Ta, VPD, and standardized precipitation-evapotranspiration index (SPEI; Vicente-Serrano et al., 2010) using statistical models. We hypothesize that: i) recent declines in tree growth are primarily driven by increasing water limitation, with stronger effects observed in Norway spruce, ii) Scots pine and Norway spruce exhibit divergent iWUE responses to increasing atmospheric demand and water stress, reflecting species-specific strategies for coping with hydrological change, and iii) tree growth and physiological responses diverge between mineral and organic soils, reflecting contrasting conditions of water availability, aeration, and nutrient dynamics.

Our specific research questions were:

- 1) How has increasing water limitation affected regional growth trends of Scots pine and Norway spruce on mineral and organic soils in southern Finland over recent decades?
- 2) What are the species-specific physiological responses of these species to contrasting water availability conditions in this region?
  - 3) How Ta, VPD and SPEI relate to growth (BAI) and physiological traits ( $\Delta^{13}$ C, iWUE) in these species within this region?

#### 2. Methods

## 2.1. Study area

## 120 2.1.1. Land and forest cover

This study was conducted in southern Finland (Figure 1). This region includes 9 provinces (Uusimaa, Southwest Finland, Satakunta, Kanta-Häme, Pirkanmaa, Päijät-Häme, Kymenlaakso, South Karelia, and South Savo) covering an area of 75,222 km². According to data from the 13<sup>th</sup> Finnish National Forest Inventory (NFI-13, 2019-2023) (Korhonen et al., 2024), 53,073 km² (71%) of the total region area is forestry land. This land is primarily privately owned and managed, with stands dominated by Scots pine (49%) and Norway spruce (36%). These tree species play a crucial role in both ecological functions and as a source of raw material for forestry sector (Henttonen et al., 2024). Notably, on mineral soils, Scots pine dominated forests cover a larger area than those dominated by Norway spruce (20,558 versus 16,137 km²) (Korhonen et al., 2024). In contrast, on organic soils, Norway spruce dominated forests are more prevalent, covering 5,298 km², compared to 4,694 km² for Scots pine dominated forests. It is important to note that a significant proportion of the organic soils with Norway spruce and Scots pine have been drained (47% and 85%, respectively).

## 2.1.2. Regional climate

The 0.5° gridded Climatic Research Unit (CRU) Time-series (TS) data version 4.08 data (Harris et al., 2020) for air temperature (Ta, °C), precipitation (P, mm), vapor pressure (VP<sub>air</sub>, in kPa), and potential evapotranspiration (PET, mm day<sup>-1</sup>) on a daily basis over the period 1990–2023 was obtained using the Climate Explorer of the Royal Netherlands

Meteorological Institute (KNMI) (Trouet and Oldenborgh, 2013). Vapor pressure deficit (VPD, kPa) was estimated using Ta, VP<sub>air</sub>, and saturation vapor pressure (VP<sub>sat</sub>, kPa), which was estimated using Eq. (1).

$$VP_{sat} = (0.6108 \times e^{((17.27 \times Ta)/(Ta+237.3))})$$
 Eq. (1)

**Figure 1. Regional climate and multi-decadal trends in southern Finland.** Data based on CRU TS data (version 4.08) for the period 1980–2021. Panels show A) mean summer temperature (Ta) for June–August (JJA), B) mean August standardized precipitation–evapotranspiration index (SPEI), C) mean JJA vapor pressure deficit (VPD), D–F) Mann-Kendall trends in JJA Ta (°C yr<sup>-1</sup>), August SPEI (unitless yr<sup>-1</sup>), and JJA VPD (kPa yr<sup>-1</sup>), respectively. Circles in panels A-C) show the location of the study sites, while the polygon outlines in panels D-F) corresponds to the southern Finland region as defined by the Finnish NFI. Panels G–I) show the regional averaged time series (1980–2021) of Ta, SPEI, and VPD, with 30-year spline curves and the shaded area shows two standard deviations around the period mean.

Finally, the VPD was estimated by subtracting VP<sub>air</sub> from VP<sub>sat</sub>. The PET was converted to mm month<sup>-1</sup> to compute the water balance needed for the estimation of the SPEI. The water balance was computed as the difference between monthly precipitation and monthly PET. A 3-month SPEI was then computed from the water balance time series using the R package *SPEI* (Vicente-Serrano et al., 2010). The 3-month SPEI was selected because it captures short to medium term moisture variability relevant to tree physiological processes during the growing season. Data processing was performed in R software using the *ncdf4* package for handling NetCDF data.

In this study, we selected Ta, VPD, and SPEI as climatic variables based on their potential impact in tree growth and iWUE during the study period, as they reflect both water availability and atmospheric demand (Babst et al., 2019; Girardin et al., 2016). The spatial distribution of Ta, SPEI, and VPD across southern Finland, along with their long-term temporal trends, reveal temperature increases of up to 0.05 C yr<sup>-1</sup> in certain areas (Figure 1D, G). Concurrently, the negative trend in SPEI indicates a decline in water availability (Figure 1E, H). In addition, VPD exhibits a similar upward trend, reflecting increased atmospheric water demand (Figure 1F, I).

# 2.2. Finnish National Forest Inventory data

GIs data were estimated to provide a broader regional context and validate site-specific observations. The GI describes the annual variation in tree growth increment that is not caused by time-dependent tree or stand factors affecting growth but by the variation of environmental factors, such as climate variability (Henttonen, 2000). In practice, age-related growth trends and other long-term developmental changes are removed by standardizing the tree growth time series before calculating the GI, so size or age are removed from the index values. This means that fluctuations in the GI reflect year-to-year changes in temperature, precipitation, soil moisture, and other environmental factors, rather than inherent age or site related growth trends. An annual GI value is typically expressed as a relative deviation from the average of the annual GIs over a long period, which is set at a value of 100 (Henttonen, 2000).

In this study, we estimated GIs separately for Scots pine and Norway spruce growing on mineral and drained organic soils in Southern Finland. Specifically, the estimation of these GIs was based on sample plot data from five consecutive Finnish NFIs, covering the period from the 9<sup>th</sup> NFI (NFI-9, 1996–2003) to the 13<sup>th</sup> NFI (NFI-13, 2019–2023). The data selection focused on southern Finland, comprising the southernmost sampling area in NFI (Korhonen et al., 2021, 2024; Tomppo et al., 2011). Radial increments, measured from tree ring increment cores from NFI temporary sample plots, were used in the GI estimation. Tree-ring cores were collected at diameter at the breast height (DBH, 1.3 m above the ground), and the ring widths from these samples were measured in the laboratory. Data were limited to trees from the dominant crown layer. To avoid confounding effects of forest management, samples were restricted to sites with no felling in the past five years and no drainage operations on mineral soils within the past 30 years. GIs were then estimated using mixed linear models (Henttonen, 2000), consistent with methods applied in the Finnish NFI (e.g., Korhonen et. al 2024).

## 2.3. Tree-ring study sites

# 2.3.1. Site selection and plot categorization

Three experimental sites per tree species (Scots pine and Norway spruce) were established in southern Finland (Figure 1) within representative forest stands. The Scots pine sites were identified as 'Solböle', 'Vastamäki', and 'Ruotsinkylä', while the Norway spruce sites were identified as 'Janakkala', 'Ränskälänkorpi', and 'Lapinjärvi'. To validate the robustness and representativeness of our sampling design, we also tested tree growth correlations between the plot-level and regional NFI data (Figure S2).

At each site, two plots (10-m radius) were established under contrasting water availability conditions, which were classified as 'dry' and 'wet', resulting in a total of 12 plots across southern Finland. This moisture categorization was based on a combination of site-specific observations. Specifically, the Scots pine plots were in different topographical contexts on mineral soils with varying susceptibility to water limitation. Plots at each site were selected using the cartographic depth-towater (DTW) index maps (2 × 2 m resolution) with 0.5 ha threshold (Murphy et al., 2007), where values below 100 indicate wet conditions and high-water retention capacity. Thus, the 'dry' and 'wet' plots were situated on forest stands on upland and lowland soils, respectively. In the case of Norway spruce plots, they were in organic soils with distinct drainage systems. Plots at each site were selected according to the ditch spacing, with the narrower the distance between ditches, the deeper the

water table level and the lower the soil moisture availability (Leppä et al., 2020a).

It should be noted that the selection of sites for Scots pine on mineral soil and Norway spruce on drained organic soil was

It should be noted that the selection of sites for Scots pine on mineral soil and Norway spruce on drained organic soil was based on the dominance of the species on these soil types in southern Finland (see subsection 2.1.).

#### 2.3.2. Soil characteristics

Soil bulk density (BD, g cm<sup>-3</sup>) was measured in both Scots pine and Norway spruce plots at 0–10 cm and 10–20 cm depths on three randomly selected locations within the plot. in Scots pine plots samples were collected using a stainless-steel cylinder (5.5 cm diameter, 4.0 cm height). In Norway spruce plots samples were collected using a stainless-steel box sampler (100 × 6.3 × 3.9 cm; length × width × height). All soil samples were dried to constant mass at 105 °C for 48 h and BD was then determined by dividing the oven-dried soil mass by the original sample volume.

Soil samples were collected next to each BD sampling point at the same depths. In Scots pine plots, samples from the organic and mineral layers across the three locations were combined into one composite sample per layer. In Norway spruce plots, all samples were pooled into a single composite sample per plot. Total carbon (C) and nitrogen (N) concentrations (%) were measured using a TruMac CN analyser (LECO Corporation, Saint-Joseph, MI, USA).

The volume of stones (Vs, %) was estimated in the 0-30 cm only in mineral soil layer at 32 sampling locations within each Scots pine plot, following 8 transects radiating from the plot centre along the cardinal directions with 4 points per transect. Vs was then assessed using a rod penetration method (Viro, 1952). Additionally, the depth of the organic layer (O<sub>depth</sub>, cm) was measured at each of these sampling locations.

# 2.3.3. Site-level climate

For each study site, we obtained the mean daily Ta and VP<sub>air</sub> from the observational data interpolated to a 10 km × 10 km grid covering Finland obtained from the Finnish Meteorological Institute (FMI). The mean daily VPD was then estimated from Ta and VP<sub>air</sub> using Eq. (1). In addition, the 3-month SPEI was obtained from the 0.5° gridded regional SPEI dataset (see subsection 2.1.2).

## 2.3.4. Tree increment core collection

Within each plot, we recorded the coordinates, DBH, total tree height (Ht, defined as the vertical distance from the base of the tree at ground level to the top of the highest living part of the crown), and tree crown base height (Hcb, defined as the vertical distance from the base of the tree at ground level to the lowest live branch) for every tree. The canopy composition was categorized by strata, quantifying the proportion of overstory, mid-story, and understory. For tree-ring analyses, 19–21 dominant and codominant trees per plot with DBH ≥ 20 cm were selected. When the required number of suitable trees was not available within the plot, additional individuals from a 5-meter buffer zone surrounding the plot were included, excluding trees near drainage ditches in Norway spruce plots. From each selected tree, one 5-mm diameter increment core was extracted at DBH using an increment borer. Cores were oriented toward the pith to optimize analysis of annual tree-ring width and carbon isotope composition (δ<sup>13</sup>C).

## 2.3.5. Tree-ring preparation and analysis

230

Tree cores were prepared and analysed following standard dendrochronological procedures (Stokes and Smiley, 1996). Each tree core was immersed in water for at least 30 minutes to avoid breaking the cores, when mounted on wooden supports, and prepared the core surface with a microtome (WSL Core Microtome; Swiss Federal Research Institute WSL, Switzerland) until the ring boundaries and wood cells were clearly visible under a binocular microscope. Then, each tree core was scanned at a resolution of 800 dpi using a colour scanner (Epson perfection v700 Photo). Finally, ring widths were measured with a precision of 0.001 mm using the WinDENDRO<sup>TM</sup> software (version Reg 2022a; Regent Instruments Inc., Quebec, Canada).

Cross-dating was performed to corroborate the calendar year assigned to each tree ring. Using standard techniques ensured the accurate assignment of calendar years to each growth ring (Stokes and Smiley, 1996). This process involved comparing ring width patterns among all samples to identify and correct any dating errors, thereby enhancing the reliability of the chronological series (Figures S3–S14 and Tables S1–12).

After cross-dating, the ring-width time series were standardized and detrended to remove age-related growth trends and emphasize the environmental and climatic signals influencing growth variability (Cook, 1987). We utilized the R package *dplR* (Bunn, 2008, 2010) for data processing and analysis. Specifically, Regional Curve Standardization (RCS) was applied to the ring width data to construct site-specific chronologies that account for variations in growth related to tree age and size (Briffa and Melvin, 2011). This method involves creating a regional growth curve by averaging ring widths across all trees of the same cambial age, allowing us to preserve to some extent the low-frequency variability related to climatic factors and

260

account for the age trends (Briffa and Melvin, 2011). Additionally, basal area increment (BAI, mm<sup>2</sup> tree<sup>-1</sup> yr<sup>-1</sup>) was estimated using the annual diameter increment per tree, using the R package *dplR* (Bunn, 2008, 2010).

# 2.3.6. Carbon isotope analysis ( $\delta^{13}C$ )

The δ<sup>13</sup>C analysis was performed on whole-ring samples to capture the integrated physiological signal of each annual ring. α-cellulose was extracted from the tree ring annual samples using standard chemical procedures adapted from Leavitt and Danzer (1993). From each plot, we selected five trees out of the 20 collected. The annual rings corresponding to each year were carefully sliced into small particles. To obtain sufficient and balanced masses for isotope analysis and reduce individual tree variability, the samples from the five trees were pooled for each annual ring, ensuring equal mass contribution from each tree (Belmecheri et al., 2022).

The pooled wood samples were placed on a Soxhlet with ethanol for 24 h to remove extractives. This step removes resins, oils, and other soluble substances that could interfere with the cellulose extraction. Following this, the samples underwent delignification by treatment with a sodium chlorite (NaClO<sub>2</sub>) solution, buffered with acetic acid (CH<sub>3</sub>COOH) to maintain a pH of around 4. This step was performed at 70°C for 6 h to degrade lignin components and other secondary metabolites in the wood sample (Tarvo et al., 2010).

The delignified samples were then treated with a 17% sodium hydroxide (NaOH) solution at room temperature for one hour to isolate α-cellulose by removing hemicelluloses (Green, 1963). After each chemical treatment, samples were thoroughly rinsed with deionized water until neutral pH was achieved. Finally, the purified α-cellulose was stabilized by treating it with a 10% acetic acid solution to neutralize any remaining alkali, and then it was rinsed again with deionized water. The samples were freeze-dried and homogenized using an ultrasonic device prior to isotope analysis. α-cellulose was used in this study because it is a structural polymer closely derived from sugars produced during the growing season, and extracting cellulose allows for more effective homogenization when pooling samples from multiple trees, improving comparability and reliability (Belmecheri et al., 2022).

The  $\delta^{13}$ C of the  $\alpha$ -cellulose samples were measured using an isotope ratio mass spectrometer (IRMS) at the Stable Isotope Laboratory (SILL) at the Natural Resources Institute Finland (Luke). Approximately 0.3 mg of each sample was weighed into tin capsules and combusted in an elemental analyser (EA) coupled to the IRMS. The results were normalized against IAEA-CH7 (-32.15‰) and in-house lactose and sucrose reference materials (-24.66‰ and -12.22‰, respectively), and are reported relative to the Vienna Pee Dee Belemnite (VPDB) standard. Repeat measurement of IAEA-CH3 material, run concurrently with the samples and normalized with the same scheme returns  $\delta^{13}$ C value of -24.77±0.03‰ (mean ± standard deviation, n = 18) in line with the certified value for this reference (-24.72‰).

# 265 2.3.7. Intrinsic water-use efficiency (iWUE)

The  $\delta^{13}$ C can be used to estimate iWUE at a variety of temporal resolution, from season (Tang et al., 2022) to multi-decadal and at spatial (leaf level to ecosystems) scales (Battipaglia et al., 2013; Gagen et al., 2011; Seibt et al., 2008). At the leaf level, the variability of  $\delta^{13}$ C in plant material represents the balance between net photosynthetic rate (A) and stomatal conductance to water vapor (gs), which both processes discriminate against  $^{13}$ C. This isotopic effect and coupling is mainly influenced by environmental factors such as soil water availability, atmospheric water demand, photosynthetically active radiation, and atmospheric CO<sub>2</sub> concentrations (Belmecheri et al., 2021; Farquhar et al., 1982; Leavitt et al., 2011; Ma et al., 2023). In this study, we describe photosynthetic carbon discrimination ( $\Delta^{13}$ C) using a simplified mechanistic model that accounts for the primary factors affecting the discrimination against  $^{13}$ C during photosynthesis (Eqs. (2 and 3)).

$$\Delta^{13}C = \frac{\delta^{13}C \text{ atmosphere} - \delta^{13}C \text{ cellulose}}{(1+\delta^{13}C \text{ cellulose})/1000}$$
Eq. (2)

Where ' $\delta^{13}$ C atmosphere' is the isotopic composition from the well mixed atmospheric CO<sub>2</sub> (Belmecheri and Lavergne, 2020) and ' $\delta^{13}$ C cellulose' is the measured isotopic composition in cellulose from the tree rings, then  $\Delta^{13}$ C expression is:

$$\Delta^{13}C = a + (b - a)\frac{c_i}{c_a}$$
 Eq. (3)

Where 'a' is the isotopic fractionation during CO<sub>2</sub> diffusion through stomata (~4.4‰), 'b' is the tissue-specific isotopic fractionation associated with net fractionation caused by Rubisco and PEPC (phosphoenolpyruvate carboxylase) and post-photosynthetic fractionation factors (Cernusak and Ubierna, 2022; Ubierna et al., 2022) integrated as (~25‰), and 'c<sub>i</sub>/c<sub>a</sub>' is the ratio of intercellular to ambient CO<sub>2</sub> concentrations (data from (Belmecheri and Lavergne, 2020). The c<sub>i</sub>/c<sub>a</sub> ratio is indicative of the relation between CO<sub>2</sub> supply to the leaves (via stomata) and CO<sub>2</sub> demand by photosynthesis in the chloroplast. At a given water evaporative demand determined by VPD, the c<sub>i</sub>/c<sub>a</sub> ratio is linked to iWUE as follows (Eqs. (4 and 5)):

$$c_i = \frac{\Delta^{13}C - a}{b - a} \times c_a$$
 Eq. (4)

$$iWUE = \frac{A}{gs} \approx \frac{gs_c*(c_a-c_i)}{gs_w} \approx \frac{c_a-c_i}{1.6}$$
 Eq. (5)

Where 'A' is the assimilation rate,  $gs_c*(c_a-c_i)$ , and ' $gs_c$ ,  $gs_w$ ' are the stomatal conductance of  $CO_2$  and water (Ehleringer and Cerling, 1995; Farquhar et al., 1982). The iWUE represents the efficiency of carbon gain per unit of potential water loss

320

325

determined by gs, and often varies with VPD, as higher VPD can lead to reduced stomatal conductance to minimize water loss (Medlyn et al., 2017).

#### 2.4. Statistical methods

# 2.4.1. Detection of changes in the time series of regional- and site-level parameters

The time series derived from the NFI GI and site-level BAI, Δ¹³C and iWUE were subjected to change-point detection using an algorithm based on the Pruned Exact Linear Time (PELT) method using the R package *changepoint* (Killick, 2011) This procedure identifies one or more time points at which the mean and variance of each time series undergo significant changes. For each time series in which one or more change points were identified, the corresponding time points were then used as initial estimates for the locations of structural breaks in a segmented regression model. This segmented regression framework enables the estimation of distinct temporal trends within different time series phases, thereby capturing shifts in the underlying dynamics. The time series was not further partitioned in cases where no significant change points were detected. This two-step approach was applied individually to each time series, resulting in a collection of models that collectively describe the temporal structure of the data, highlighting those chronologies with and without detected shifts.

# 2.4.2. Sensitivity of regional- and site-level parameters to environmental stressors

To evaluate the influence of the low and high frequency climate variability on inter-annual tree growth and physiology, we use both the detrended and not detrended NFI-based GI, tree-ring variables (BAI and Δ¹³C), and the monthly climate indices (Ta, VPD, and SPEI). Detrending was performed using the super smoother Friedman method implemented in the dplR package (Bunn, 2010). For each site and month, we then fit a simple linear model of the form of: (variable (detrended and not) ~ Climate index (detrended and not)). We applied this simple linear model for all combinations of sites, species, and months including previous fall and extracted the coefficient of determination (R²), slopes, and p-values from each model run.

Additionally, we used linear mixed-effects models (LMMs) to examine the sensitivity of tree growth and physiological responses to SPEI and VPD. Ta was excluded from the models due to its high collinearity with VPD. The analysis was conducted on BAI and  $\Delta^{13}$ C chronologies, with monthly SPEI and VPD averaged across the growing season (i.e., May-July (MJJ)). Climate predictors and years were centred before modelling to facilitate interpretation of changes in intercepts and differences in predictor effects, which correspond to interaction effects across periods. A categorical variable representing pre-2015 and post-2015 periods was included to detect shifts in climate sensitivity associated with intensified water limitation after 2015, identified post-hoc through changes in mean variance and trends (subsection 2.4.1). Models were fitted separately for Scots pine and Norway spruce using the following structure:

$$y_i = \alpha_{\text{period}(i)} + \beta_{\text{period}(i)} \cdot \text{SPEI}_i + \gamma_{\text{period}(i)} \cdot VPD_i + e_{\text{site}(i)} + \varepsilon_i$$
 Eq. (6)

Where  $y_i$  represents either BAI or  $\Delta^{13}$ C of the i'th observation,  $e_{\text{site}}$  is a site-specific random intercept, and  $\varepsilon_i$  is a random residual, which is assumed uncorrelated between observations. Because annual observations within a plot are temporally dependent, we modelled within-plot residuals with a first-order autoregressive process, AR (1), indexed by year this allows residuals closer in time to be more correlated and improves precision of fixed-effect estimates under temporal dependence. Models were fitted using restricted maximum likelihood (REML). Parameters  $\alpha_{\text{period}}$ ,  $\beta_{\text{period}}$ , and  $\gamma_{\text{period}}$ , for period  $\in$  {pre-2015,post-2015}, as well as the between-site variances of random effect e and the AR(1) parameters were estimated using the lme function of the R package nlme (Pinheiro et al., 1999).

#### 335 **3. Results**

#### 3.1 NFI-based tree growth patterns in southern Finland

Our analysis of the NFI-based GI data for Scots pine and Norway spruce on both mineral and organic soils during the period 1990–2022 provides a regional overview on tree growth variability modulated by environmental conditions across southern

Figure 2. National Forest Inventory (NFI) based tree growth patterns in southern Finland. Panels show NFI-based standardized growth index (GI) for Scots pine (A) and Norway spruce (B) from 1990 to 2022 on mineral (red) and organic (green) soils. The upper panels illustrate the annual variability in GI for each tree species and soil type, with thin lines depicting the mean annual values, while thick lines showing the 10-year smoothing splines to illustrate long-term trends. The lower panels show the number of trees used to derive each annual GI estimate for each tree species and soil type.

Finland (Figure 2). Specifically, Scots pine shows consistent GI values on both soil types during the early years of the study period. However, a small decline in growth from 2005 onwards in trees growing in organic soils, whereas a more gradual reduction was observed in trees on mineral soils beginning around 2010. In contrast, Norway spruce exhibited greater interannual variability in GI across both soil types. Notably, after a period of steady growth increase around 2015, Norway spruce showed a marked decline on both mineral and organic soils, with the reduction being slightly more pronounced on mineral soils.

#### 3.2. Site-level patterns in tree growth and iWUE

Sample plots in Scots pine sites on mineral soils exhibited contrasting topographical conditions, with upland dry plots showing significantly higher depth-to-water index (257±78 cm) and a greater proportion of rocks (55.3±3.0%) than the lowland wet plots, which had values of 36±4 cm and 10.7±6.1%, respectively. In contrast, plots in Norway spruce sites on organic soils were characterized by distinct drainage systems, with wet plots showing significantly wider ditch spacing (65±16 m) compared to dry plots (46±4 m). Despite these topographical and hydrological disparities, plot-level measurements revealed no significant differences in forest stand and soil characteristics between dry and wet plots within each tree species (Tables 1 and 2).

Table 1. Forest stand characteristics for Scots pine and Norway spruce sites under 'Dry' and 'Wet' conditions. Td: tree density (trees ha<sup>-1</sup>), DBH: diameter at breast height (cm), Ht: total tree height (m), Hbc: tree crown base height (m), BA: basal area (m<sup>2</sup> ha<sup>-1</sup>), Age: mean stand age (years). The values represent the mean±standard deviation. For each tree species and variable, a Bonferroni-adjusted post hoc test was applied to compare differences in means between plots (p < 0.05). No significant differences were identified between the 'Dry' and 'Wet' plots for any variable across tree species. n = 3 sites per tree species and plot type.

| Tree species  | Plot | Td           | DBH      | Ht       | Hbc      | BA              | Age     |
|---------------|------|--------------|----------|----------|----------|-----------------|---------|
|               |      | (trees ha-1) | (cm)     | (m)      | (m)      | $(m^2 ha^{-1})$ | (years) |
| Scots pine    | Dry  | 743±81       | 19.1±3.8 | 16.1±3   | 9.4±2.8  | 25.1±4.6        | 81±12   |
|               | Wet  | 860±139      | 21.3±4.6 | 19.2±3.5 | 12.2±3.4 | 36.3±8.2        | 74±31   |
| Norway spruce | Dry  | 785±296      | 21.7±4.3 | 19.3±2.8 | 8.3±2.5  | 30.3±0.3        | 83±36   |
|               | Wet  | 859±292      | 22.5±3.9 | 20.5±2.5 | 9.3±1.5  | 36.6±5.1        | 77±4    |

Table 2. Soil characteristics at Scots pine and Norway spruce sites under 'Dry' and 'Wet' conditions. BD: soil bulk density (g cm<sup>-3</sup>), C: carbon concentration (%), N: nitrogen concentration (%), C/N: carbon/nitrogen ratio,  $O_{depth}$ : topsoil organic layer depth (cm), Vs: volume of stones up to 30 cm depth (%). BD values refer to the topsoil organic layer and the mineral soil layer (0–20 cm) for Scots pine plots, and to the organic soil layer (0–20 cm) for Norway spruce plots. C, N, and C/N values are reported for the mineral soil layer (0–20 cm) for Scots pine plots, and for the organic soil layer (0–20 cm) for Norway spruce plots.  $O_{depth}$  and Vs are provided only for Scots pine plots; these are not applicable for Norway spruce plots, where soils are organic and stone content is zero. The values represent the mean  $\pm$  standard deviation. For each tree species and variable, a Bonferroni-adjusted post hoc test was applied to compare differences in means between plots (p 

However, Vs differed significantly between plot types in Scots pine, as indicated by different letters. n = 3 sites per tree species and plot type.

| Tree species  | Plot | BD              | С          | N             | C/N      | Odepth  | Vs                    |
|---------------|------|-----------------|------------|---------------|----------|---------|-----------------------|
|               |      | $(g cm^{-3})$   | (%)        | (%)           |          | (cm)    | (%).                  |
| Scots pine    | Dry  | $0.73\pm0.07$   | 4.65±2.39  | 0.16±0.07     | 27.2±4.9 | 5.4±0.5 | 55.3±3.0 <sup>b</sup> |
|               | Wet  | $0.74 \pm 0.04$ | 2.53±0.92  | $0.11\pm0.04$ | 23.6±1.0 | 7.4±1.7 | 10.7±6.1 <sup>a</sup> |
| Norway spruce | Dry  | $0.13\pm0.03$   | 49.10±0.81 | 1.73±0.29     | 29.1±5.5 | _       | _                     |
|               | Wet  | $0.12 \pm 0.03$ | 49.30±1.23 | $1.79\pm0.14$ | 27.7±2.2 | _       | _                     |

From around 2010-15 onward, Norway spruce displayed a reduction in basal area growth, especially under dry conditions (Figure 3B) making the growth plateaued and decline after 2015. Site-level data indicate that BAI often increased until 2005–2010 before declining, with dry plots showing earlier and more pronounced reductions (Figures S15–S16). In contrast, Scots pine maintained stable growth, particularly under wet conditions. Although some dry condition pine plots showed declines after 2010, in general pine stands showed minimal long-term changes (Figure 3).

When these growth patterns are compared with  $\Delta^{13}$ C and its derived iWUE, Norway spruce typically showed smaller iWUE increases, along with more pronounced growth decline (Figure 3D). Scots pine, however, with a more stable growth, exhibited a marked overall increase in iWUE and a slight extra increase after 2015, suggesting stronger response via stomatal regulation under increased atmospheric demand and reduced water availability (Figure 4). Temporal variability at individual sites is shown in more in Figures S15–S16.

Figure 3. Trends in basal area increment (BAI) and intrinsic water use efficiency (iWUE) for Scots pine and Norway spruce under 'Dry' and 'Wet' conditions during 1990-2022. Panels A-B) show values for BAI, while panels C-D) show values for iWUE. Solid and orange lines display dry conditions, while dashed and blue lines represent wet conditions. Shaded areas indicate the 90% confidence intervals around the smoothed lines using a 30-year smoothing spline.

# 385 3.3. Change-points in tree growth and iWUE over the study period

Segmented regression analysis (Figure 4) revealed that most change-points in GI, BAI,  $\Delta^{13}$ C, and iWUE occurred between 2010 and 2015. Following these change points, Norway Spruce showed a clear downward trend for GI, BAI and  $\Delta^{13}$ C, most notable in BAI but also reflected in NFI-based GI data. Scots pine also showed significant trend changes in both BAI and GI, though these were more pronounced at sites with drier conditions.

Figure 4. Change points in tree growth and water use over the study period 1990-2022. Panels show the time series for A-B) standardized Growth Index (GI) from the Finnish National Forest Inventory (NFI), C-D) basal area increment (BAI), E-F), carbon isotope discrimination ( $\Delta^{13}$ C), and G-H) intrinsic water-use efficiency (iWUE) for Scots pine (left panels) and Norway spruce (right panels). Within each panel, the grey lines represent time series with no detected change points. In contrast, the coloured lines and dashed lines highlight time series in which the segmented regression method detected one or more change points. Below each time-series panel is a density plot illustrating the temporal distribution of estimated change points across all time-series over the probability density function (PDF). The *y*-axis differs between plots because the PDF is normalized, but the maximum density (area below the function) depends on the number of change points for each variable in each dataset.

A consistent pattern in  $\Delta^{13}$ C and iWUE was the synchronous detection of change-points around 2015 across both species. However, the magnitude of iWUE responses diverged: Scots pine exhibited a greater number of significant change-points and a steeper post-2015 increase in iWUE. In contrast, Norway spruce showed only modest shifts in iWUE, despite more substantial growth declines under increasingly dry climatic conditions. Further information on the change-point detection results is provided in Table S13.

## 3.4. Environmental influence on tree growth and carbon isotope discrimination

Linear regression analysis (Figure 5) illustrates how 3-month seasonal means of key climate variables, including Ta, SPEI, and VPD, relate to tree-ring characteristics and NFI-based GIs. Among these, water availability (SPEI) and atmospheric demand (VPD) emerged as the strongest influential drivers. Ta showed similar temporal patterns to VPD, suggesting its effects are mediated through increased evaporative demand.

At the site-level,  $\Delta^{13}$ C showed the strongest relationships with climatic variables during July-August, when low SPEI and high VPD coincided with reduced carbon isotope discrimination (Figure 5 A-C). These patterns suggest that intensified summer water stress induces greater stomatal closure, particularly in Scots pine, leading to lower  $\Delta^{13}$ C values. Ta exerted a weaker but occasionally notable influence on  $\Delta^{13}$ C, due to its indirect role in increasing VPD. This indicates that high summer Ta can further exacerbate water stress, most noticeably in Scots pine sites. In contrast, BAI but especially NFI-based GI exhibited weaker correlations with moisture and temperature variables compared to  $\Delta^{13}$ C (Figure 5 D-I).

Figure 5. Heatmaps showing the strength and significance of relationships between climate variables and tree growth and physiological indicators across different sites and seasonal windows (3-month periods). Panels illustrate the relationships between Ta, SPEI, and VPD with  $\Delta^{13}$ C (A-C), BAI (D-F), and regionally NFI-based GI (G-I). In all panels, the *x*-axes represent the 3-month periods from September of previous year to November to the current growth year, while the *y*-axes show site-level chronologies for Scots pine and Norway spruce (panels A-F), as well as regional chronologies (panels G-I). Colour shading represents the coefficient of determination (R<sup>2</sup>) from simple linear regressions, ranging from white (low R<sup>2</sup>) to red (high R<sup>2</sup>). Positive (+) and negative (-) signs indicate the direction and statistical significance of the relationship (p 

Figure 6. Sensitivity of basal area increment (BAI) and carbon isotope discrimination ( $\Delta^{13}$ C) to recent changes in water limitation. Standardized effect sizes (slopes  $\pm$  95% confidence intervals) of summer (mean May–July SPEI and VPD) on annual tree ring BAI (A, C) and  $\Delta^{13}$ C (B, D) for Scots pine and Norway spruce respectively based on the lmm model runs described in Eq. (6). Blue and red points represent the pre-2015 and post-2015 periods, respectively, capturing shifts in climate sensitivity associated with recent intensified water limitation. Slopes indicate the estimated change in BAI (m² ha⁻¹ yr⁻¹) and  $\Delta^{13}$ C (‰) per unit change in the respective climate variable. Error bars denote 95% confidence intervals. Effects are considered statistically significant when confidence intervals do not overlap zero.

BAI in Scots pine shows minimal changes in intercepts across periods (Table 3). Growth sensitivities to SPEI remain close to zero for SPEI and are negative for VPD (Figure 6a). In contrast,  $\Delta^{13}$ C shows clearer and more consistent climate signals, with positive SPEI slopes, and negative slopes related to VPD effect on  $\Delta^{13}$ C (Figure 6b) showing no significant

difference between periods (Table 3). Together, these results indicate that while growth responses in Scots pine remain limited by water, the isotopic sensitivity to moisture stress has persisted in the post-2015 period.

In Norway spruce, significant changes in intercepts across periods supports the different baselines in BAI and  $\Delta^{13}$ C between periods (Figure 4 and Table 3). VPD emerges as a clear constraint on BAI after 2015 (Figure 6c), although during the pre-2015 the VPD slope is not significantly different from zero and the difference between periods shows to be marginally significant (Table 3). The SPEI slope is significantly negative in pre-2015 and decreases significantly after 2015 (Figure 6c and Table 3). Additionally, Norway spruce  $\Delta^{13}$ C, sensitivity to VPD and SPEI show no significant differences between periods.

Under the May-July climate window, SPEI and VPD sensitivity strengthens more clearly in Norway spruce BAI after 2015, indicating enhanced growth limitation due to these environmental factors. In contrast, Scots pine Δ<sup>13</sup>C shows more prominent and consistent climate sensitivity than Norway spruce Δ<sup>13</sup>C, with positive responses to SPEI and negative responses to VPD. Scots pine BAI, however, remains comparatively buffered from climatic trends.

Table 3. Pre- and post-2015 changes in the sensitivity of the basal area increment (BAI) and carbon isotope discrimination (Δ<sup>13</sup>C) to climate variables for Scots pine and Norway spruce. Changes in model-estimated parameters for the Standardized Precipitation-Evapotranspiration Index (SPEI, β) and vapor pressure deficit (VPD, γ) for the period May-July (MJJ) based on model runs described in Eq. (6) (p < 0.001 "\*\*\*", 0.01 "\*\*\*", 0.05,"\*", 0.1 ".").

|               |                                      | BA       | AI .     | $\Delta^{13}$ | <sup>3</sup> C |
|---------------|--------------------------------------|----------|----------|---------------|----------------|
| Species       | Parameter                            | estimate | p        | estimate      | p              |
| Scots pine    | $lpha_{ m pre}$                      | 0.435    | ***      | 19.391        | ***            |
|               | $\alpha_{ m post} - \alpha_{ m pre}$ | 0.055    | 0.064.   | -0.016        | 0.897          |
|               | $oldsymbol{eta}_{ m pre}$            | 0.002    | 0.787    | 0.266         | 0.001***       |
|               | $eta_{ m post} - eta_{ m pre}$       | -0.018   | 0.363    | 0.109         | 0.479          |
|               | $\gamma_{ m pre}$                    | -0.456   | ***      | -4.956        | ***            |
|               | $\gamma_{ m post} - \gamma_{ m pre}$ | 0.127    | 0.434    | 0.521         | 0.659          |
| Norway spruce | $lpha_{ m pre}$                      | 0.518    | ***      | 20.167        | ***            |
|               | $\alpha_{ m post} - \alpha_{ m pre}$ | 0.162    | 0.001*** | 0.594         | ***            |
|               | $oldsymbol{eta}_{ m pre}$            | -0.034   | 0.004**  | 0.135         | 0.003**        |
|               | $eta_{ m post} - eta_{ m pre}$       | -0.058   | 0.033**  | 0.06          | 0.562          |
|               | $\gamma_{ m pre}$                    | -0.1     | 0.529    | -1.438        | 0.019**        |
|               | $\gamma_{ m post} - \gamma_{ m pre}$ | -0.385   | 0.079.   | 0.795         | 0.34           |

## 4. Discussion

In this study, we used regional GIs from the NFI and site-level tree-ring analyses on BAI and  $\Delta^{13}$ C to evaluate how Scots pine and Norway spruce respond to intensifying water stress across mineral and organic soils under contrasting water-availability conditions in southern Finland. The findings highlight that both species exhibit growth declines under increasing VPD and decreasing SPEI, yet Norway spruce shows a more pronounced reduction in growth and lower long-term response in iWUE compared to Scots pine. The contrasting responses between these two species to ongoing shifts in atmospheric moisture demand (Figure 1) have significant implications for future boreal forest dynamics, particularly in drought-prone environments such as southern Finland (Figure 1). Rising atmospheric dryness can induce considerable stress on trees, primarily through increased evaporative demand and the potential for greater water loss (Adams et al., 2017; Frank et al., 2015). Thus, the ability of different species to cope with water limitations will likely determine their future growth, survival, competitive success and management strategies (Levesque et al., 2013). In recent years, Scots pine showed greater growth reductions on organic soils, whereas Norway spruce exhibited more pronounced relative declines on mineral soils (Figure 2). These divergent patterns indicate that future forest vulnerability under climate change will be shaped not only by species-specific physiological sensitivities but also by the spatial heterogeneity of soil types across the boreal landscape in southern Finland.

# 4.1. Environmental drivers on recent growth declines

The NFI data show a clear decline in GIs for both Scots pine and Norway spruce over the most recent decade (Henttonen et al., 2024) (Figure 2). These records are consistent with our independent records from plot-based growth measurements (Figure S2). The fact that both species show growth declines across mineral and organic soils suggests that broader environmental factors are driving these observed growth patterns.

According to our observations, increased VPD is a significant driver of the recent growth decline observed in Norway spruce in southern Finland. Based on site-level data, Scots pine showed a slight decrease in growth in dry conditions during the last decade. However, in both dry and wet conditions, growth variability was accompanied by a steep increase in iWUE (Figure 3). In contrast, Norway spruce exhibited a substantial decline in growth in both dry and wet conditions in our study sites, especially after 2015, with a rather long term stable iWUE likely related to the lower stomatal sensitivity, characteristic to the species (Ewers et al., 2001; Oren et al., 1999).

Declining SPEI values and increasing VPD intensify physiological stress by exerting additional pressure from drier soils and a heightened atmospheric moisture demand (Allen et al., 2010, 2015; Williams et al., 2013). This process accelerates soil moisture depletion and increases drought severity (Novick et al., 2016). The impact of these processes is evident in the responses documented in Scots pine and Norway spruce (Figure 6). Norway spruce showed a notable decline in growth, particularly after 2015, and was more severely impacted by rising VPD, this response is likely attributed to its physiological

characteristics and adaptation to colder, more humid conditions, resulting in a sharp decline in growth due to increasing water stress (Lagergren and Lindroth, 2002; Levesque et al., 2013; Pichler and Oberhuber, 2007; Thum et al., 2008).

Overall, our results align with broader trends observed not only in boreal forests across Fennoscandia (Henttonen et al., 2024; Junttila et al., 2024; Laudon et al., 2024), but also throughout Europe (Knutzen et al., 2025; Römer et al., 2025), where soil water limitation and increasing VPD have been linked to increased tree mortality and growth declines (Adams et al., 2017; Grossiord et al., 2020). Recent studies further indicate that the degree of atmospheric drying registered over the last 400 years is unprecedented (Treydte et al., 2023), making these observations a regional, emerging problem that is largely driven by ongoing climate change. This intensification of water stress not only heightens the vulnerability of these forest ecosystems but also poses a severe threat to tree species with limited adaptability to arid conditions (IPCC, 2023).

# 4.2. $\Delta^{13}$ C reveals tree responses to environmental stress

The analysis of  $\Delta^{13}$ C in tree rings has been shown to offer valuable insights into the physiological adjustments of trees in response to environmental stress (Brooks et al., 1998; McDowell et al., 2003).  $\Delta^{13}$ C reflects the ratio of intercellular to atmospheric CO<sub>2</sub> concentrations, which in turn is influenced by stomatal conductance and photosynthetic rate (Farquhar et al., 1982). Under water-limited conditions, trees close their stomata to conserve water, but stomatal regulation can vary among species, resulting in interspecies discrepancies in reduced stomatal conductance and limited CO<sub>2</sub> uptake. This water-carbon balance, in turn, leads to changes in the  $\Delta^{13}$ C values fixed within the tree ring (Barbour and Farquhar, 2000; Francey and Farquhar, 1982).

Moreover, the observed changes in  $\Delta^{13}$ C values are indicative of adjustments in iWUE (Ehleringer et al., 1993) and carbon assimilation under variable water availability. An increase in iWUE suggests that trees optimize water use by regulating stomatal aperture to balance water loss with carbon gain (Walker et al., 2020). The strong linear relationship found between  $\Delta^{13}$ C and moisture-related climate variables, even when using detrended time series, suggests that trees respond physiologically to the interannual variations in water availability and not only to the long-term trends (Figure 5 and S17).

Although detrended BAI and GI data may not show strong correlations with climate variables due to different limiting factors affecting growth (Cabon et al., 2020; Cuny et al., 2015), the physiological responses captured by Δ<sup>13</sup>C provide evidence of water stress impacting the long-term growth trends (Figures 4, 5, and 6). This observation highlights an apparent contradiction, where growth data alone may under detect the impacts of water related physiological stress, emphasizing the importance of integrating physiological indicators (such as Δ<sup>13</sup>C) with growth measurements to achieve a more comprehensive understanding of tree responses.

Moreover, our findings align with those of McDowell et al. (2010), who demonstrated that drought-induced mortality was related to a significant decline in growth alongside low iWUE sensitivity to water limitation. Here, 'iWUE sensitivity' refers to the magnitude of the temporal trend or slope in iWUE, where a significant sensitivity indicates a steep trend, and a low sensitivity indicates a nearly flat slope (Figure 3D). McDowell et al. (2010) observed that surviving trees experiencing water stress exhibited increased stomatal closure, leading to reduced carbon assimilation and growth, and predisposing them to

higher mortality risk. However, trees that died showed a larger growth decline but lower sensitivity in iWUE. Consequently, a sharp decline in growth coupled with relatively constant iWUE, as observed in Norway spruce relative to Scots pine, suggests that Norway spruce has lower stomatal regulation (i.e., a low water regulation buffer) during periods of increasing water demand. The corresponding abrupt decline in growth is consistent with the dual pattern of growth and  $\Delta^{13}$ C observed elsewhere (McDowell et al., 2010; Wang et al., 2021), suggesting that Norway spruce may be experiencing similar physiological stress that could increase its vulnerability to mortality under continued water-limiting conditions.

Norway spruce showed more stable  $\Delta^{13}$ C values over time than Scots pine, with the latter showing declining values in dry conditions indicating consistent stomatal regulation to buffer water loss. Moreover, the observed stability in Scots pine growth suggests an effective water-use strategy, enabling it to sustain growth under a long trend of declining water availability, thus avoiding abrupt changes in secondary growth. The findings indicate that Scots pine shows resilience to water limitations, maintaining growth and carbon assimilation (as seen in BAI and GI) while regulating water loss through increased iWUE.

The variations in both  $\Delta^{13}$ C and iWUE of the two species underscore their divergent strategies for coping with increasing water limitations. The use of  $\Delta^{13}$ C as a physiological indicator helps to our understanding of how trees respond to climatic variability from a mechanistic angle. The value of  $\Delta^{13}$ C in interpreting tree responses to environmental stress has been demonstrated in previous studies (Brooks et al., 1998; Saurer et al., 2014), reinforcing its applicability in the present research.

#### 4.3. Species and soil influences on tree water stress and forest management implications

Norway spruce showed stronger growth sensitivity and weaker iWUE response than Scots pine under increasing water limitation in southern Finland, highlighting important considerations for the management of these tree species in this boreal region. However, the contrasting responses of both species across different soil types also carry significant implications for forest management. Specifically, Scots pine experienced greater growth reductions on organic soils, likely due to poor aeration and reduced water availability during dry periods. Although organic soils generally retain more moisture, they can develop hydrophobic surface layers and root-soil hydraulic disconnection during extended drought, limiting infiltration and uptake and effectively, disconnecting trees from near-surface water (Doerr et al., 2000; Hökkä et al., 2025). In contrast, Norway spruce exhibited more pronounced growth declines on mineral soils, which typically retain less moisture. Spruce's shallow rooting system in adverse conditions further limits its ability to access water on these sites, increasing its vulnerability to summer drought (Kalliokoski et al., 2008; Puhe, 2003). These findings highlight the need for forest management strategies that consider both species-specific physiological traits and edaphic conditions when assessing drought risk and planning adaptive silvicultural interventions.

Focusing on contrasting soil moisture conditions, specifically, lowland versus upland positions on mineral soils for Scots pine, and poorly versus well-drained organic soils for Norway spruce, we found that each species exhibited site-dependent responses to water limitation. Scots pine growing in upland mineral soils exhibited reduced BAI but a notable increase in

iWUE, indicating a physiological shift toward tighter stomatal control under greater water stress. Conversely, Scots pine on lowland mineral soils displayed higher BAI but less pronounced increases in iWUE, reflecting more favourable water availability that supports productivity but with less stomatal regulation under drought stress. These divergent patterns underscore the critical role of soil moisture regimes in mediating species responses to water limitation (Bose et al., 2020; Green et al., 2019). From a forest management perspective, upland forest stands may benefit from interventions such as reducing stand density, enhancing structural diversity through varied tree ages and sizes, or promoting mixed-species composition to alleviate water stress and enhance resilience (Aldea et al., 2024; Pardos et al., 2021) In contrast, lowland forest stands could be managed to optimize growth while closely monitoring soil moisture conditions for emerging drought risks. Integrating these spatially explicit strategies will be essential for sustaining forest productivity and stability under intensifying climate change in southern Finland.

Norway spruce growing on both well- and poorly drained organic soils showed marked declines in growth coupled with only modest increases in iWUE, highlighting a limited ability to cope with intensified water stress. Consequently, current forest management approaches relying heavily on the prevalence of Norway spruce in drought-prone peatland areas of southern Finland may need to be reconsidered. Promoting coniferous mixed-species stands could enhance ecosystem resilience, while strategies aimed at reducing drainage intensity and increasing water retention in peat soils may mitigate drought impacts (Hökkä et al., 2025; Pardos et al., 2021). It is also imperative to explore practical management practices in drained organic soils, including adjusting thinning regimes and/or promoting alternative final harvesting methods, such as continuous-cover forestry, to reduce intra-stand competition for water (A. Lehtonen, Eyvindson et al., 2023; Leppä, Hökkä, et al., 2020; Leppä, Korkiakoski, et al., 2020; Tikkasalo et al., 2023). Additionally, modifying drainage systems through block ditching and/or rewetting can enhance water storage (Laine et al., 2011), while monitoring of soil conditions is critical to inform adaptive silvicultural decisions (Mäkipää et al., 2024). These approaches collectively offer promising pathways to improve tree performance and ecosystem stability under changing hydrological conditions.

#### 5. Conclusions

Our findings demonstrate that forests in southern Finland are increasingly affected by water limitations, particularly over the past decade. Norway spruce showed a marked post-2015 decline in growth, while Scots pine exhibited only minor growth reductions beginning around 2010. In parallel, iWUE increased more steeply in Scots pine than in Norway spruce, suggesting a weaker physiological adjustment in spruce and reinforcing its greater vulnerability to rising water limitation and prolonged drought stress. Interannual variability in both growth and iWUE closely followed changes in SPEI and VPD, highlighting the pivotal role of water limitation in driving recent physiological responses and growth declines. Linear mixed-effects models further confirm a post-2015 increase in VPD and SPEI sensitivity for Norway spruce growth, whereas Scots pine maintained relatively buffered growth and consistent  $\Delta^{13}$ C responses.

Our findings also demonstrate that tree responses to increasing water limitation are strongly modulated by soil type and site-level moisture conditions. Scots pine exhibited greater drought vulnerability on organic soils, due to limited aeration and impaired rooting conditions during dry periods, while Norway spruce showed more pronounced growth reductions on mineral soils with lower moisture retention capacity. Notably, contrasting species responses were evident within soil types:

Scots pine was more sensitive to water stress on upland than on lowland mineral soils, whereas Norway spruce responded more strongly on well-drained than on poorly drained organic soils. Together, these patterns underscore the importance of both soil moisture retention and rooting conditions in shaping species-specific sensitivity to drought.

Taken together, our findings have important implications for boreal forest management under climate change. Integrating species-specific physiological responses and soil-mediated influences on tree water stress into adaptive forest management strategies will be essential to support the long-term resilience and productivity in boreal forests facing increasingly warm and

water-limited climate.

#### Data and code availability

BAI,  $\Delta^{13}$ C, and iWUE data analysed in the Stable Isotope Laboratory (SILL) of the Natural Resources Institute Finland (Luke), along with GI data from the Finnish NFI, and the R code generated in this study, are openly available in the Zenodo digital repository at <a href="https://doi.org/XXXXXXXX">https://doi.org/XXXXXXXXX</a>

## 580 Author contributions

Paul Szejner: Conceptualization, Formal analysis, Investigation, Methodology, Visualization, Writing - original draft & editing.

Eduardo Martínez-García: Conceptualization, Formal analysis, Investigation, Methodology, Data curation, Writing - review & editing

Helena Haakana: Data curation, Methodology, Writing - review & editing

Juha Heikkinen: Formal analysis, Writing - review & editing

Katja T. Rinne-Garmston: Writing - review & editing

Petri Kilpeläinen: Data curation, Writing - review & editing

Giles Young: Data curation, Writing - review & editing

Elina Sahlstedt: Data curation, Methodology, Writing - review & editing

Raisa Mäkipää: Writing - review & editing

Aleksi Lehtonen: Conceptualization, Project administration, Resources, Writing - review & editing

# **Competing interests**

The authors declare that they have no known competing financial interests or personal relationships that could have appeared to influence the work reported in this paper.

## Acknowledgements

We also thank the field and laboratory staff at the Natural Resources Institute Finland (Jari Ilomäki, Ari Ryynänen, Petri Salovaara, and Kati Tammela) for their assistance in field data collection and tree-core analysis.

# Financial support

This study was supported by the Natural Resources Institute Finland (Luke) and the Ministry of Agriculture and Forestry in Finland (Meka 2.0, grant n° VN/10860/2023-MMM-2) and the European Union's Horizon 2020 (Forwards, grant n° 101084481).

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
