# Peer review of "Norway spruce shows stronger growth sensitivity and weaker intrinsic water-use efficiency response than Scots pine under increasing water limitation in southern Finland"

_EGUsphere, 2025_

## Referee Comment (RC1)

This study compares growth trajectories and water use efficiency between Norway spruce and Scots pine. It provides compelling evidence that Scots pine has maintained a relatively constant growth despite increasing water limitations due to an increased water use efficiency, while Norway spruce has been less capable of adjusting its water use efficiency and thereby suffered growth declines. Furthermore, the study makes use of NFI data to successfully extrapolate the results to a larger scale. The study is highly relevant for both future research and to forest management as it provides important information on how the two most dominant tree species (in the region) compare in relation to climatic variables that may change drastically in the future. The study seems to be very well performed, and the results provide a good basis for the conclusions drawn. I have only minor comments where I mainly would like to see clarifications (especially in the methods section), and a few open questions on how some of the results have been found.

**Specific comments**

Line 107-117: Both hypotheses and research questions are stated. Both are clear, concise and relate well to what the reader can expect from the paper. However, maybe one (either hypotheses or questions) would suffice? This is of course just a matter of taste, and not of scientific importance, but the text would likely be made even more concise by choosing only one of them.

Equation 1: Here, perhaps you could just show the entire equation for VPD rather than VPsat (since VPD is the variable used throughout the analyses)? You also refer to this equation on line 205 as an equation for calculating VPD. I suggest writing it as "VPD = (0.6108 x e((17.27xTa)/(Ta+237.3)) – VPa", and explain that the formula in parentheses refer to VPsat. Alternatively adjusting the text on line 205 to clarify that Eq. (1) does not calculate VPD.

Line 153-170: This section is somewhat confusing regarding detrending of data. It seems here like an intrinsic part of GI is the fact that the data has been standardized to remove time-dependent factors. However, in your statistical analyses you explain that GI data is used both as detrended and non-detrended. Is the GI simply radial increments (as stated in the 2nd paragraph), in which case it is directly comparable to the data collected from the six experimental sites? Or is it already manipulated, in which case detrending the GI might be a case of "double-detrending" of data? This could be clarified, perhaps through a thorough explanation of how GI is calculated.

Line 175: The correlation is between plot-level BAI and NFI GI. It is not entirely clear from this section if the correlations were carried out on detrended or non-detrended data. This could be added for clarity.

Line 224: There are some (although very few) series that have low correlations to the master chronology. Were these removed prior to analyses or kept and trusted? The low number of low correlations mean that they likely do not influence the results much, but a note on how they have been handled could be added for transparency.

Regarding the Carbon isotope analysis: I am unfamiliar with methods regarding this, and can thereby not criticize the method. However, the analysis seems to have been carried out by a reliable laboratory and I see no reason to think that the data is of poor quality.

Line 313: According to the results you seem to have done 3-month averages rather than "each site and month" which seems to be only presented in supplementary results. This could be clarified.

Line 319: Here you use MJJ while you present climatic trends of JJA (or August) in Figure 1. Is there a reason why MJJ was chosen? Furthermore, coherency between Figure 1 and this choice could provide the reader with a better understanding of the input to the analysis.

Figure 2: There is a drastic decline in data towards the later years (as shown in the lower panel) – Why is that and doesn't it risk skewing your interpretation of the temporal development of growth? If you agree that the decreasing data quantity could influence the results, I suggest adding a caveat in the text about this.

Table 1 and 2: The post-hoc tests that the significance levels are based on don't seem to be mentioned in the methods. These could be added so that they don't appear in the results for the first time.
Further, I personally do not agree with using p-value adjustments as this can potentially force a Type II error onto data that otherwise reveal clear differences. However, this is a personal opinion, and p-value corrections are very common, so I do not expect you to change this – I just wanted to air my opinion on the matter.

Line 375: You write that the decline of Norway spruce BAI was especially large under dry conditions. Considering Figure 3B, I'm not sure I agree. The decline seems to be the same from 2015 in both soil moisture types. If anything, the relative decline seems greater under wet conditions (e.g. by looking at the mean average lines: 0.4/0.8 = 0.5 versus 0.6/1.0 = 0.6, when comparing the last data entry to that of 2015). If you still think that the decline is larger under dry conditions, the reasoning behind this could be clarified.

Line 388: Regarding the GI and drier conditions. You mention here that trend changes in GI were more pronounced in drier conditions. However, can the comparison between drained organic soils and mineral soils be considered as a comparison between "wet" and "dry" soils?
You introduce the soil types in the paragraph around L88-L100. However, it is not clear from this that organic soils (drained or not) should be considered wetter than mineral soils. I suggest clarifying that the trend changes were more pronounced on mineral soils – not at sites with drier conditions.

Line 388: Regarding the BAI and drier conditions. According to Table S13, only 2 of 6 time series changed significantly, and 1 of those was from a wet site (not counting the wet site with a significant change in slope but without a positive-to-negative trend change). While the change in the drier site indeed was more negative, the total slope change was greater in the wet site. Therefore, while I think you can argue for the fact that the drier sites show more significant trend changes, you could also argue for the opposite. Hence, I would like to see a clarification on what you base that "BAI [...] were more pronounced at sites with drier conditions" on.

Figure 4 (figure text): "In contrast, the coloured lines and dashed lines highlight time series in which the segmented regression method detected one or more change points." – It is not clear from the text what the difference is between the coloured solid and coloured dashed lines are. I suggest clarifying (if I have understood it correctly) that the coloured solid lines represent time series with one or more change points detected and that coloured dashed lines represent the model output from the segmented regression analysis.

Line 396-407: It is not clear whether these results are based on detrended or non-detrended data. In the methods you mention that analyses are carried out on both. This could be clarified, even if the results are similar independent of detrending.

Line 466: "Norway spruce […] was more severely impacted by rising VPD" – this can pe inferred from the analysis of pre- and post-2015. However, the overall regression analyses seem to complicate the findings somewhat. Except for previous year autumn VPD, there doesn't seem to be much that suggest that Norway spruce is more sensitive to VPD (in fact, current year summer months' VPD suggest a weakly opposite pattern). This complexity could be elaborated in this section.

Line 473: "Recent studies further indicate that the degree of atmospheric drying registered over the last 400 years is unprecedented" – This makes it seem as though the entire 400 years are unprecedented. I suggest rephrasing to something like: "Recent studies further indicate that the current degree of atmospheric drying is unprecedented compared to the last 400 years"

Line 538: This part of the discussion is very nicely written and highly relevant to forest management, and most of the suggestions seem intuitive. However, I'm not sure I understand why structural diversity would lead to alleviated water stress. Could this be explained or expanded on?

Line 551: Same here. It is not intuitive to me why CCF would reduce intra-stand competition for water.

Line 549: Why is this only imperative in drained organic soils? The paragraph starts by showing that well- and poorly drained soils both have marked declines in growth. And considering Figure 2B, it seems like the mineral soil, rather than the drained organic soil, is more sensitive.

**Suggestions on spelling and sentence structure:**
In **BOLD** are suggestions for added words,
 are suggestions for removed words.

Line 19: "increased more steeply **in** Scots pine"

Line 41: "These droughts, often coinciding with elevated air temperatures (Ta), increase**d** vapor pressure deficit (VPD) and reduce**d** soil moisture, are likely to become more frequent under continued warming"

Line 74: "reflects a species**'** ability" – apostrophe.

Line 90: "Drainage induced lowering **of** the water table"

Line 117: "How **does** Ta, VPD and SPEI"

Line 126: "raw material for **the** forestry sector"

Line 149: "0.05 **°**C" – degree sign

Line 166: "Tree-ring cores were collected at  breast height"

Line 191: "the plot. **In** Scots pine" – capital letter

Line 204: "we obtained the mean daily Ta and VPair from the observational data interpolated to a 10 km × 10 km grid covering Finland  from the Finnish Meteorological Institute"

Line 219: "Each tree core was immersed in water for at least 30 minutes to avoid breaking the cores, when mounted on wooden supports, and  the core surface **was prepared** with a microtome"

Line 301: "(Killick, 2011)**.** This" – dot

Line 310-316: Present and past tense: "we use" and "we fit" compared with "Detrending was" and "We applied".

Line 375: "making the growth plateau and decline"

Line 383: "is shown in more **detail** in Figures S15–S16."

Line 398: "[...] y climate variables,  Ta, SPEI, and VPD" – "including" suggests that there are other variables as well.

Line 412: "Growth sensitivities  remain close to zero for SPEI and are negative for VPD"

Line 466: "Norway spruce showed a notable decline in growth, particularly after 2015, and was more severely impacted by rising VPD**. This** response is likely attributed to its physiological characteristics and adaptation to colder, more humid conditions, resulting in a sharp decline in growth due to increasing water stress"

Line 540: "**.** In contrast, [...]" – dot

Line 566: "Scots pine exhibited greater drought vulnerability on organic soils, **likely** due to limited aeration and impaired rooting conditions during dry periods" – as it is now, it sounds as though the mechanistic pathway of impaired rooting conditions was tested. I suggest adding some cautious wording.